

# Genomic insights into *Lactobacillus gasseri* and *Lactobacillus paragasseri*

Adriana Ene[1], Natalie Stegman[1], Alan Wolfe[2] and Catherine Putonti[1,2,3]

[1] Bioinformatics Program, Loyola University of Chicago, Chicago, IL, United States of America
[2] Department of Microbiology and Immunology, Loyola University of Chicago, Maywood, IL, United States of America
[3] Department of Biology, Loyola University of Chicago, Chicago, IL, United States of America

## ABSTRACT

**Background**. Antimicrobial and antifungal species are essential members of the healthy human microbiota. Several different species of lactobacilli that naturally inhabit the human body have been explored for their probiotic capabilities including strains of the species *Lactobacillus gasseri*. However, *L. gasseri* (identified by 16S rRNA gene sequencing) has been associated with urogenital symptoms. Recently a new sister taxon of *L. gasseri* was described: *L. paragasseri*. *L. paragasseri* is also posited to have probiotic qualities.

**Methods**. Here, we present a genomic investigation of all ($n = 79$) publicly available genome assemblies for both species. These strains include isolates from the vaginal tract, gastrointestinal tract, urinary tract, oral cavity, wounds, and lungs.

**Results**. The two species cannot be distinguished from short-read sequencing of the 16S rRNA as the full-length gene sequences differ only by two nucleotides. Based upon average nucleotide identity (ANI), we identified 20 strains deposited as *L. gasseri* that are in fact representatives of *L. paragasseri*. Investigation of the genic content of the strains of these two species suggests recent divergence and/or frequent gene exchange between the two species. The genomes frequently harbored intact prophage sequences, including prophages identified in strains of both species. To further explore the antimicrobial potential associated with both species, genome assemblies were examined for biosynthetic gene clusters. Gassericin T and S were identified in 46 of the genome assemblies, with all *L. paragasseri* strains including one or both bacteriocins. This suggests that the properties once ascribed to *L. gasseri* may better represent the *L. paragasseri* species.

# INTRODUCTION

*Lactobacillus* species are a common colonizers of the human microbiome, including the gastrointestinal (GI) tract, urinary tract, and vaginal microbiota (*Ravel et al., 2011*; *Heeney, Gareau & Marco, 2018*; *Fok et al., 2018*). One member of this genus, *Lactobacillus gasseri*, has been the focus of studies associated with weight loss and its probiotic benefits (*Crovesy et al., 2017*; *Oh et al., 2020*; *Sun et al., 2020*). Furthermore, *L. gasseri* has been shown to mitigate *Helicobacter pylori* infection and ameliorate diarrhea (*Selle & Klaenhammer, 2013*).

Corresponding author
Catherine Putonti, cputonti@luc.edu

*L. gasseri* is an important constituent of the female urogenital tract (*Ravel et al., 2011*; *Fok et al., 2018*). It is able to prevent other bacteria from growing in the same environment, protecting the host from pathogens (*Selle & Klaenhammer, 2013*). *L. gasseri* is an effective antimicrobial (*Spurbeck & Arvidson, 2010*; *Scillato et al., 2021*) and antifungal (*Matsuda et al., 2018*; *Parolin et al., 2021*) of urogenital pathogens. In the vaginal microbiota, *L. gasseri* can be a dominant member in healthy women without bacterial vaginosis (BV), and thus a marker for vaginal health (*Srinivasan et al., 2012*). However, in women with human papillomavirus infections, *L. gasseri* is frequently isolated. (*Gao et al., 2013*). Within the urinary tract, *L. gasseri* has been associated with urgency urinary incontinence (UUI) in females (*Pearce et al., 2014*), although it also is frequently isolated from the bladders of continent females (*Price et al., 2020*).

Whole genome sequencing efforts of isolates from the human microbiota led to the identification of two distinct subgroups of *L. gasseri* (*Tada et al., 2017*), which later led to the classification of *L. gasseri*'s sister taxon: *Lactobacillus paragasseri*. First described in 2018 (*Tanizawa et al., 2018*), little is known about the species. One *L. paragasseri* strain from the GI tract has been found to inhibit the *Lactobacillus* species *L. iners* (*Nilsen et al., 2020*), and *L. paragasseri* strains have been examined for their potential probiotic use (*Mehra & Viswanathan, 2021*; *Shiraishi et al., 2021*). Recently, *L. paragasseri* was posited as a resilient member of the healthy urinary microbiota (*Ksiezarek et al., 2021*). The species does, however, have the potential for pathogenicity; *L. paragasseri* was been found to be the causative agent of a cavernosal abscess in one individual (*Toyoshima et al., 2021*).

It is important to note that the aforementioned studies associating *L. gasseri* in the urogenital microbiota and symptom status (*Srinivasan et al., 2012*; *Gao et al., 2013*; *Pearce et al., 2014*; *Price et al., 2020*) predate the discovery of *L. paragasseri*. Moreover, as prior studies note, it is difficult to distinguish *L. gasseri* from *L. paragasseri* using commonly employed techniques for typing species, namely MALDI and 16S rRNA gene sequencing (*Toyoshima et al., 2021*; *Zhou et al., 2020*). The most reliable way to distinguish the two taxa is through whole genome sequencing (*Tanizawa et al., 2018*; *Zhou et al., 2020*). Recently, whole genome sequencing of 92 *L. gasseri* and *L. paragasseri* strains from fecal samples identified genomic distinctions between the two species with regards to their CRISPR-Cas systems, bacteriocin operons, and carbohydrate-active enzymes (*Zhou et al., 2020*). Here, we present a genome analysis of all publicly available whole genome sequences of *L. gasseri* and *L. paragasseri*. This includes 79 strains isolated from the vaginal tract, GI tract, urinary tract, oral cavity, and lungs.

## MATERIALS & METHODS

The publicly available sequences of *L. gasseri* and *L. paragasseri* totaling 79 genomes were retrieved from NCBI as of June 23, 2021. Table S1 lists the sequences included in this study. First, 16S rRNA gene sequences were extracted from each genome assembly using Python and Biopython. When multiple 16S rRNA gene sequences were identified, as is the case for long-read sequencing assemblies and complete genome assemblies, all copies were included in the set of sequences. The extracted 16S rRNA sequences were manually examined through

Geneious Prime (Biomatters Ltd., Auckland, NZ) and aligned using the MAFFT v7.388 (*Katoh & Standley, 2013*) plug-in through Geneious Prime. The phylogenetic tree was derived using the FastTree 2.1.12 (*Price, Dehal & Arkin, 2010*) plug-in through Geneious Prime and visualized using iTOL v6 (*Letunic & Bork, 2016*).

Next, we estimated the average nucleotide identity (ANI) using pyani v0.2 (*Pritchard et al., 2015*). From the ANI (ANIm metric) percentage identity values, we classified the genomes into the two species using the 95% threshold (*Jain et al., 2018*).

The genomes were then examined using anvi'o v7.2 (*Eren et al., 2021*). Contigs less than 500 bp were removed from consideration using the command anvi-script-reformat-fasta. Afterwards, the command anvi-gen-contigs-database was used to perform gene calls for each contig using Prodigal generating anvi'o databases for each genome. The commands anvi-run-hmms and anvi-run-ncbi-cogs were also run to annotate the genes in the anvi'o databases. The anvi-pan-genome command was used to create the pangenome of all 79 genomes with an Markov Chain algorithm (MCL) threshold of 8. The concatenated single copy core genome was found using the command anvi-get-sequences-for-gene-clusters with the–min-num-genomes-gene-cluster-occurs 79–max-num-genes-from-each-genome 1–concatenate-gene-clusters. A phylogenetic tree, consisting of the aligned single copy core genome, was derived, and visualized as described above. Functionality for genes of interest were confirmed *via* blastp queries to the NCBI nr database.

Each genome sequence was screened for prophage sequences using PHASTER (*Arndt et al., 2016*). While PHASTER predicted intact, questionable, and incomplete prophage sequences, only intact prophage sequences were examined in depth. Homologous intact prophages were identified by clustering the nucleotide sequences using a 70% percent identity threshold. Clustering was performed using USEARCH v.11.0.667 (*Edgar, 2010*). Each cluster was manually inspected, and the cluster's sequences were aligned as described above.

Lastly, each genome was queried for secondary metabolites *via* antiSMASH using the all extra features on and the rest were the default parameters (*Blin et al., 2021*). The bacteriocin sequences found by antiSMASH were aligned using MAFFT (v7.388) (*Katoh & Standley, 2013*). Trees were derived and visualized as described above. Reference sequences were retrieved from GenBank for the gassericin T (*L. gasseri* LA327: Accession No. LC389592) and gassericin S (*L. gasseri* LA327: Accession No. LC389591) nucleotide sequences.

# RESULTS

While the 16S rRNA gene sequences of *L. gasseri* and *L. paragasseri* strains are 99.9% identical, the few SNPs present result in the two species clading separately (Fig. 1). The similarity in the 16S rRNA gene sequences observed here concurs with prior studies remarking on the inability to distinguish between the two species *via* the 16S sequence (*Toyoshima et al., 2021*; *Zhou et al., 2020*). However, the two species can be distinguished by just two nucleotide differences: position 95 (C/T) in the V1 region and 1046 (A/T) in the V6 region. Some strains include additional polymorphisms.

Next, whole genome sequences were examined. The 95% threshold for pairwise ANI values was used as the species boundary between the two sister taxa. Based upon our ANI

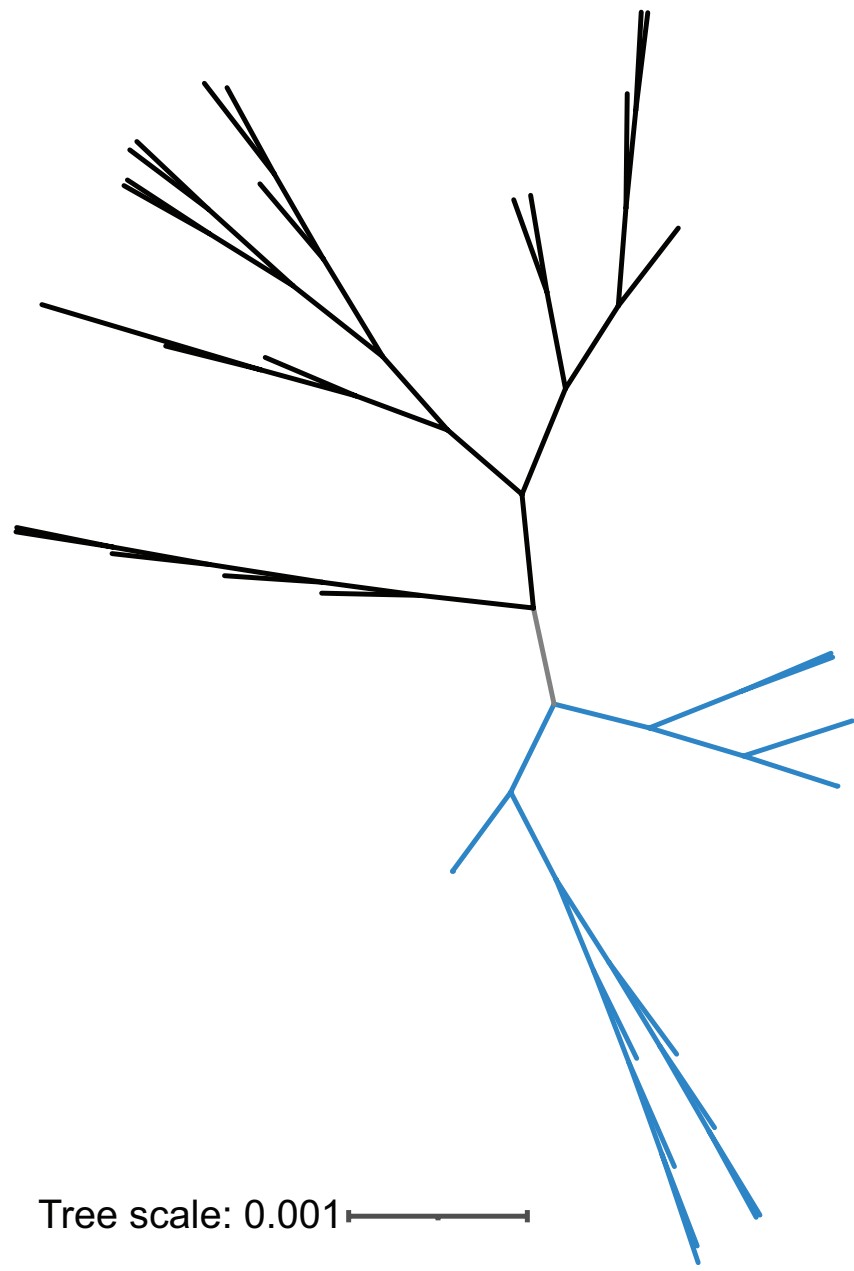

Tree scale: 0.001

**Figure 1** **16S rRNA gene tree of _L. gasseri_ (black) and _L. paragasseri_ (blue) isolates.** Species designation is based upon ANI analysis (Fig. 2).

calculations, 20 of the genomes classified in GenBank as _L. gasseri_ were determined to be _L. paragasseri_ strains. This misclassification is primarily because these genome assemblies were deposited prior to the discovery of _L. paragasseri_ in 2018. In contrast, all strains that were identified as _L. paragasseri_ in GenBank were confirmed to be _L. paragasseri_. Based upon the ANI analysis, our data set includes 40 representatives of _L. gasseri_ and 39

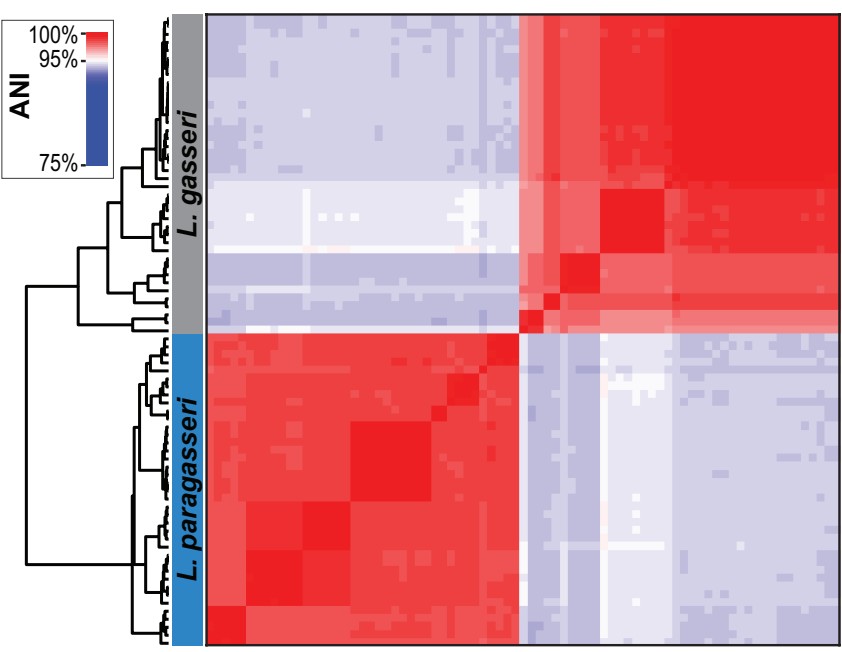

**Figure 2** **ANI analysis of *L. gasseri* and *L. paragasseri* strains.** The grey rectangle on the upper left side shows the *L. gasseri strains* while the blue rectangle below it shows the *L. paragasseri strains.*

representatives of *L. paragasseri* (Fig. 2). This strain assignment concurs with the branching of strains based upon 16S rRNA gene sequences (Fig. 1).

Next, the pangenome and set of single copy genes in the core genome of the 79 *L. gasseri* and *L. paragasseri* genomes was identified. The pangenome consists of 4,069 groups of orthologous genes. Two of these genes are conserved among all strains of *L. paragasseri* and are not present in any of the *L. gasseri* strains. These genes encode for a phosphatidylserine decarboxylase, also found in other lactobacilli, including *L. taiwanensis* and *L. johnsonii*, and beta-galactosidase, also found in *L. johnsonii*. Thus, these genes cannot serve as a *L. paragasseri*-specific gene marker as their use would not be able to distinguish between *L. paragasseri* and other lactobacilli. There are no genes that are both conserved among all strains of *L. paragasseri* and absent from the *L. gasseri* strains. The single copy core genome of all 79 *L. gasseri* and *L. paragasseri* assemblies contains 242 single copy orthologous genes. Using this core genome, the phylogenomic tree was derived (Fig. 3). Like that observed for the 16S rRNA and ANI analyses, the tree shows a clear distinction between the two species.

The source for each strain was identified from the genome metadata and associate literature (Table S1; Fig. 3). Strains of both *L. gasseri* and *L. paragasseri* have been isolated from stool/GI, vaginal, urine, lung/aspirate, and breast milk samples. Additionally, *L. paragasseri* has been isolated from a wound ($n = 1$) and the oral cavity ($n = 1$). The isolation source of several of the strains could not be determined (Fig. 3, dark gray "Source" boxes).

The genomes were next screened for secondary metabolites. Of the 79 genome assemblies examined, secondary metabolites were identified in 46 assemblies, including 7 *L. gasseri*

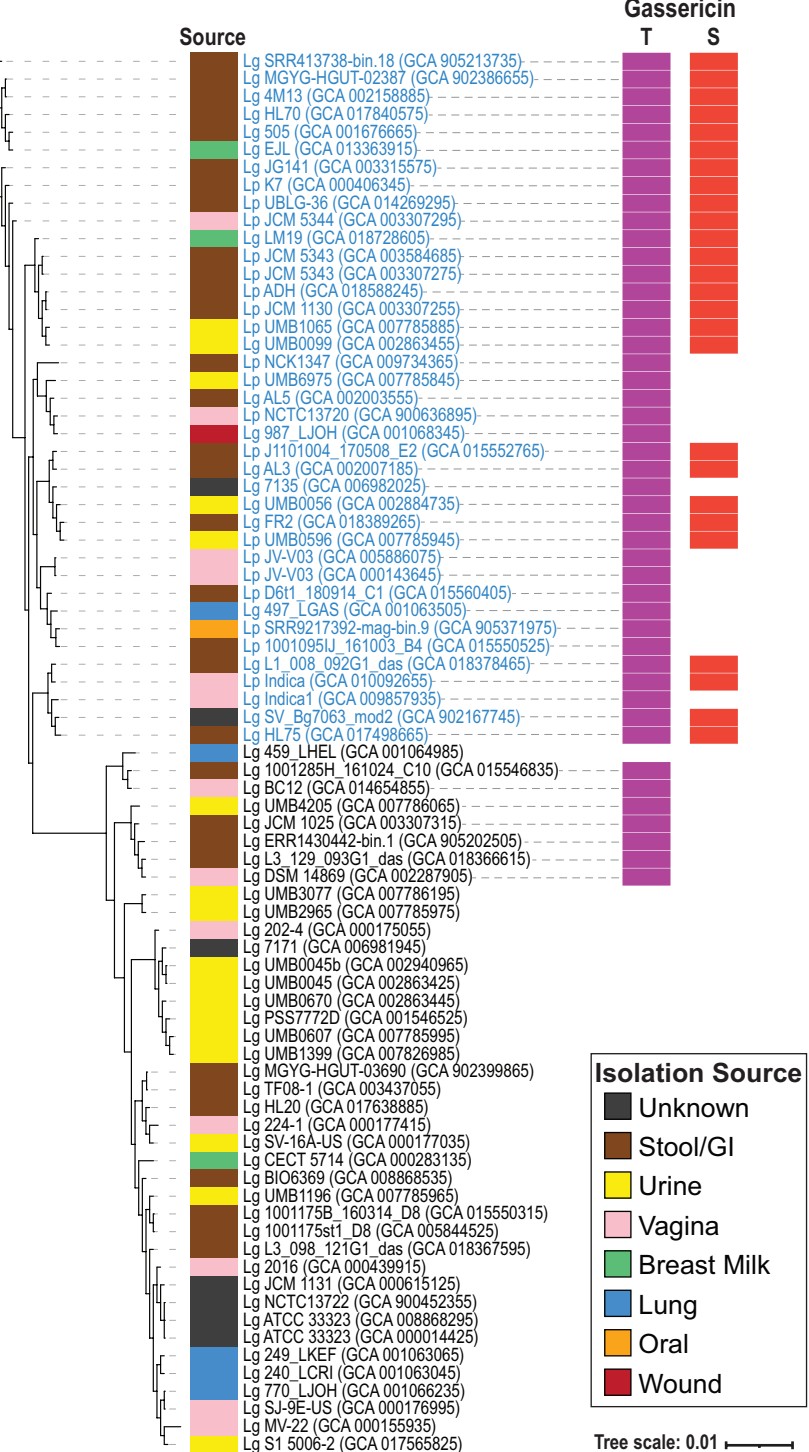

**Figure 3** *L. gasseri* **and** *L. paragasseri* **core genome phylogenetic tree.** Strains deposited in the database as *L. gasseri* are labeled as "Lg" while those deposited in the database as *L. paragasseri* are labeled as "Lp". Strains in blue font are *L. paragasseri* and black font are *L. gasseri*, according to our ANI analysis (Fig. 2). The isolation source of the genomes is indicated according to the legend. Gassericin T and S presence is indicated by the purple or red boxes, respectively.

assemblies and 39 *L. paragasseri* assemblies. The most frequent biosynthetic cluster identified were the two ribosomally synthesized and post-translationally modified peptide product (RiPP) clusters gassericin T and gassericin S. Gassericin T was identified in all *L. paragasseri* assemblies (*n* = 39) but only 7 *L. gasseri* assemblies (Fig. 3, purple boxes); *L. paragasseri* UMB1065 was predicted to contain two gassericin T clusters. The gassericin T cluster includes nine genes (MiBIG database records: BGC0000619 and BGC0001601). While the gassericin T cluster in *L. paragasseri* 497_LGAS encoded for all nine of these genes, the remaining identified gassericin T clusters lacked one or more genes. At one extreme, *L. paragasseri* UMB1065 and *L. paragasseri* UMB0596 only encode for three of the genes. On average, the gassericin T clusters identified here include seven of the nine genes (Table S2). In contrast, the gassericin S clusters, which were only identified in *L. paragasseri* assemblies (*n* = 26; Fig. 3, red boxes), were well conserved in relation to the reference sequence (MiBIG database record: BGC0001601). Twenty-five of the strains encoded for all three of the gassericin S genes in the cluster; *L. paragasseri* Indica only encoded for two of the genes (Table S2).

Next, a phylogenetic tree was derived for the nucleotide sequences of the two biosynthetic clusters. For gassericin T, full length sequences were found in all strains except for *L. paragasseri* UMB6985 (omitted from tree), which had a truncated coding sequence. The phylogenetic tree (Fig. 4A) shows two clades: one for *L. gasseri* and one for *L. paragasseri* strains. On average, the sequence similarity between these gassericin T nucleotide sequences was 84.6%. In contrast, the nucleotide sequences for the gassericin S clusters were nearly identical (average pairwise nucleotide identity = 99.7%) (Fig. 4B). In addition to the two gassericin clusters, acidocin B, furan and lactocillin were identified in 3, 7, and 3 strains, respectively (Table S2).

The *L. gasseri* and *L. paragasseri* genomes were screened for prophage sequences. In total 82 intact prophage sequences were identified. These sequences represent 38 different prophages. Sequence homology between the prophages identified and characterized strains include metagenome assembled phages as well as *Lactobacillus*-infecting characterized phages (Table S3). Twenty-four of these prophages were identified only in a single genome sequence. Of the 38 different prophages identified, 10 were only found only in a *L. gasseri* strain(s), and 19 were found only in a *L. paragasseri* stain(s). It is important to note that none of these prophages were present in all *L. gasseri* or all *L. paragasseri* strains indicative of acquisition events post-speciation. The remaining nine prophages are present in both *L. gasseri and L. paragasseri* strains.

We investigated further the prophages infectious of both species. Phylogenetic analysis found instances in which prophage sequences did (prophage "both_3"; Fig. 5A) and did not (prophage "both_1"; Fig. 5B) clade by the *Lactobacillus* species harboring it. The two examples shown in Fig. 5 both show greatest sequence similarity to *Lactobacillus* phage jlb1 (both_3 greatest pairwise sequence similarity: 81% query coverage; both_1 greatest pairwise sequence similarity: 93% query coverage). Pairwise sequence identity values can be found in Table S3. A temperate phage, *Lactobacillus* phage jlb1 has previously been shown to contribute to horizontal gene transfer (*Baugher, Durmaz & Klaenhammer, 2014*).

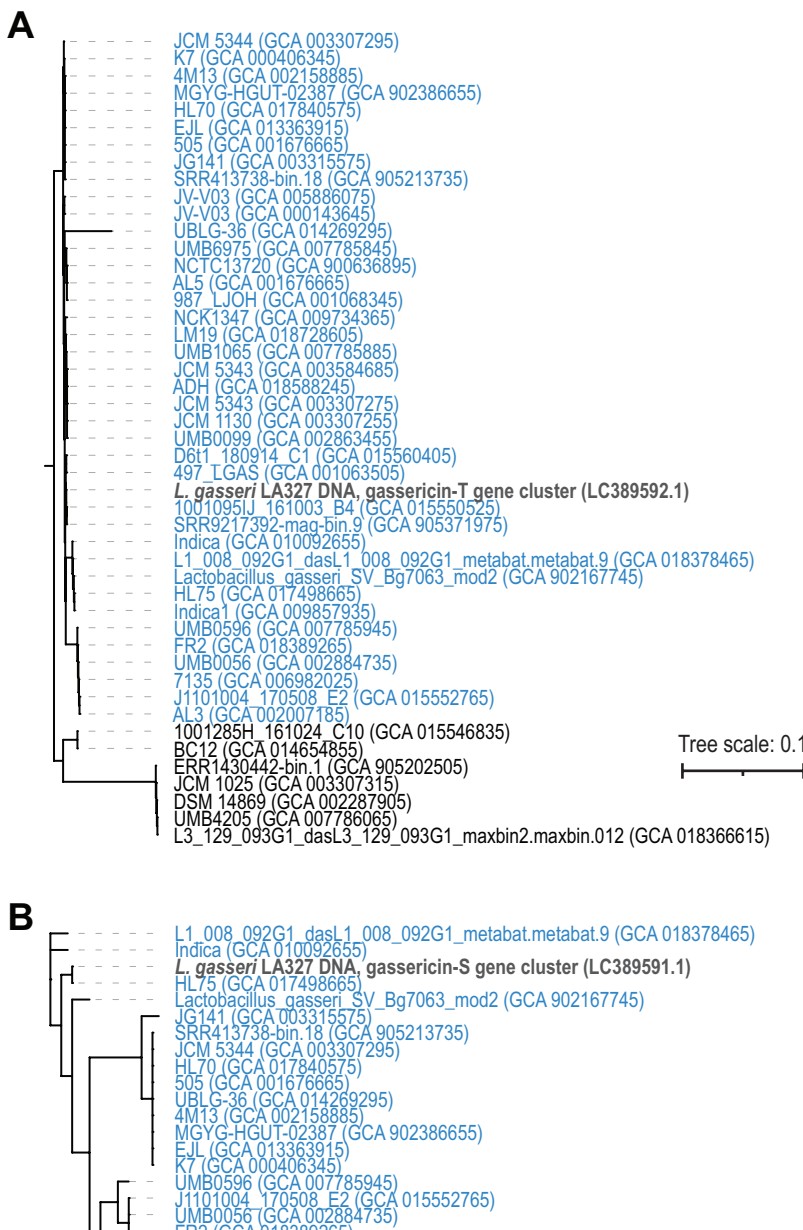

**Figure 4   Phylogenetic trees of (A) gassericin T and gassericin S (B).** *L. paragasseri* strains are in blue font and *L. gasseri* strains are in black font using the species designation determined by the ANI analysis (Fig. 2). Reference sequences for the gassericin sequences are shown in bold, gray font.

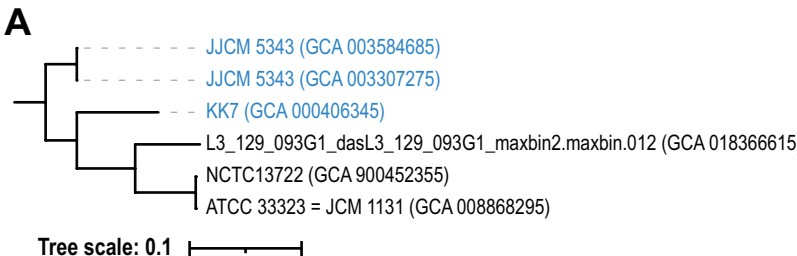

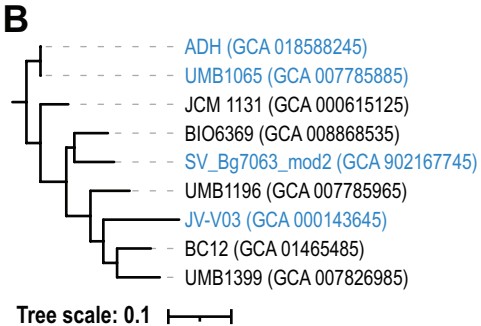

**A**

JJCM 5343 (GCA 003584685)
JJCM 5343 (GCA 003307275)
KK7 (GCA 000406345)
L3_129_093G1_dasL3_129_093G1_maxbin2.maxbin.012 (GCA 018366615)
NCTC13722 (GCA 900452355)
ATCC 33323 = JCM 1131 (GCA 008868295)

Tree scale: 0.1

**B**

ADH (GCA 018588245)
UMB1065 (GCA 007785885)
JCM 1131 (GCA 000615125)
BIO6369 (GCA 008868535)
SV_Bg7063_mod2 (GCA 902167745)
UMB1196 (GCA 007785965)
JV-V03 (GCA 000143645)
BC12 (GCA 01465485)
UMB1399 (GCA 007826985)

Tree scale: 0.1

**Figure 5** **Phylogenetic tree for two prophages—(A) group "both_3" and (B) group "both_1".** *L. paragasseri* strains are indicated in blue font while *L. gasseri* strains are in black font.

Further analysis is required to ascertain if the identified prophages also are temperate phages and if they are capable of infecting both species.

# DISCUSSION

Through our analysis of publicly available genomes of *L. gasseri* and *L. paragasseri* isolates, we found that targeting variable regions within the 16S rRNA gene is insufficient to distinguish between the two species. While 16S metagenome surveys that target the V1 or V6 regions may be able to capture the mutations, it is unlikely that bioinformatic tools would call the two species as two different operational taxonomic units (OTUs) or amplicon sequence variants (ASVs). Furthermore, urobiome studies to date have primarily relied on the V4 region as it is able to distinguish between other common community taxa (*Hoffman et al., 2021*). Targeting the V1-V3 and V3-V4 regions of the 16S rRNA gene sequence have been found to perform best for vaginal microbiome studies (*Hugerth et al., 2020*). It is worth noting that the aforementioned studies associating *L. gasseri* with vaginal or urinary and vaginal symptoms cannot resolve it from. *L. paragasseri* as they target regions of the 16S rRNA gene sequence that are identical for *L. gasseri* and *L. paragasseri* (*Srinivasan et al., 2012*; *Gao et al., 2013*; *Pearce et al., 2014*; *Price et al., 2020*).

The distinction between *L. gasseri* and *L. paragasseri*, however, can be made when examining whole genome sequences. The gene content of these two species differs, notably in the bacteriocins encoded. We hypothesize that the gassericin T gene was lost in the deeper

clade of the *L. gasseri* strains, based upon the core phylogenomic tree (Fig. 3). Gassericin T and S, which were only recently isolated from *L. gasseri* LA327 and described (*Kasuga et al., 2019*), are likely contributors to the antimicrobial activity associated with *L. paragasseri* rather than *L. gasseri*. The genome sequence for *L. gasseri* LA327 is not publicly available, but based upon our study, we hypothesize that this strain is likely a *L. paragasseri* strain.

   *L. gasseri* bacteriocins, including gassericin T, have been explored for their use as food preservatives (*Arakawa et al., 2009*). Gassericins also have been shown to inhibit the growth of pathogenic bacteria (*Itoh et al., 1995*), capable of inhibiting the invasion of competing strains or pathogens, as was shown in the case of the *L. paragasseri* K7 (*Nilsen et al., 2020*; *Shiraishi et al., 2021*). Our analysis finds that *L. paragasseri* encodes for more bacteriocins than *L. gasseri*. This concurs with a previous examination of *L. gasseri* and *L. paragasseri* genomes from the gut (*Zhou et al., 2020*). These findings suggest that the antimicrobial and antifungal properties associated with *L. gasseri* may better describe *L. paragasseri*, with prior strains misclassified as *L. gasseri*.

   We did not find a candidate gene marker to distinguish between the two species. While a 2017 study of gene content in these two species did find "group" specific genes (*Tada et al., 2017*), our analysis, which included all publicly available genomes to date, did not find any genes that are both conserved among all strains of *L. paragasseri* and absent from the *L. gasseri* strains. Furthermore, the two genes conserved among all strains of *L. gasseri* and absent from *L. paragasseri* are also found in the genomes of other lactobacilli. The gene content similarity between strains of the two species suggests that either these species have very recently diverged and/or gene exchange is frequent between the two species. Phages may have contributed and continue to contribute to the divergence of these two species through horizontal gene transfer (Table S3; Fig. 5). Furthermore, niche-specialization may be driving the differences in gene content between strains of the same species. Both *L. gasseri* and *L. paragasseri* have been isolated from very different environments (Table S1; Fig. 3). Niche-specific adaptations have previously been observed between one *L. paragasseri* strain from the GI tract and one *L. paragasseri* strain from the vagina (*Pan et al., 2020*). To associate *L. gasseri* and *L. paragasseri* with urogenital health, future studies need to include shotgun metagenomic sequencing and/or isolate whole genome sequencing.

## CONCLUSIONS

Our comparative genomic study of all 79 publicly available *L. gasseri* and *L. paragasseri* strains finds that the two species only can be reliable distinguished by whole genome sequence analysis. The gene content of strains from these two species is quite similar suggesting recent divergence and/or frequent gene exchange. The presence of prophage sequences may be contributing to the divergence as well as the observed similarity in gene content. Furthermore, bacteriocins previously ascribed to *L. gasseri* strains are actually encoded by *L. paragasseri* strains. As a result of our study, we recommend that future studies must utilize shotgun metagenomic sequencing or whole genome sequencing of isolates to definitively determine if one or both of these species are associated with urogenital symptoms.

### Funding
This work was supported through an investigator-initiated grant from the Kimberly Clark Corporation awarded to Catherine Putonti and Alan J. Wolfe. The funders had no role in study design, data collection and analysis, decision to publish, or preparation of the manuscript.

### Grant Disclosures
The following grant information was disclosed by the authors:
Kimberly Clark Corporation.

### Competing Interests
Alan J. Wolfe serves on the Scientific Advisor Boards of Pathnostics and Urobiome Therapeutics.

### Author Contributions
- Adriana Ene conceived and designed the experiments, performed the experiments, analyzed the data, prepared figures and/or tables, authored or reviewed drafts of the paper, and approved the final draft.
- Natalie Stegman performed the experiments, analyzed the data, prepared figures and/or tables, authored or reviewed drafts of the paper, and approved the final draft.
- Alan Wolfe conceived and designed the experiments, authored or reviewed drafts of the paper, and approved the final draft.
- Catherine Putonti conceived and designed the experiments, analyzed the data, prepared figures and/or tables, authored or reviewed drafts of the paper, and approved the final draft.

### Data Availability
The publicly available genome assemblies from NCBI's Assembly database are available in Table S1.

### Supplemental Information
Supplemental information for this article can be found online at http://dx.doi.org/10.7717/peerj.13479#supplemental-information.

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
