# Peer review of "Genomic insights into Lactobacillus gasseri and Lactobacillus paragasseri"

_PeerJ, doi:10.7717/peerj.13479_

## Round 0.1 · original submission · Minor Revisions

Your manuscript has been reviewed by two experts in the field, one of whom recommended it be returned for revisions. However, the other reviewer recommended rejection on the basis that many of the findings have already been reported in a paper published in 2017 by Tada et al.

It is my view that there is merit in reproducing previous results, so I will consider it for publication if you address the comments and suggestions made by reviewer 2 as well as properly reference the work by Tada et al and expand your discussion to include what is similar about your methods and findings versus what new you can add with your more recent analysis.

Reviewer 1 ·

Basic reporting

.

Experimental design

.

Validity of the findings

.

Additional comments

Most of the findings in this paper have already been described in https://www.ncbi.nlm.nih.gov/pmc/articles/PMC5633531/, in which it is shown that publicly available genomes of L. gasseri are divided into 2 distinct groups by ANI, and the strains belonging to one of the groups (later denoted as L. paragasseri in 2018) specifically have genes related to gassericin T production.

Reviewer 2 ·

Basic reporting

In their manuscript, Ene and colleagues report on a comparative genomics study including 79 publicly available genomes for Lactobacillus gasseri and Lactobacillus paragasseri. The latter has been described only recently, and the current study proves that several L. gasseri strains described and sequenced before, need to be reclassified as L. paragasseri. This study is of interest for researchers in the fields were these species are relevant. In a wider view, this study shows the strength of genome sequencing, and most likely studies as this one will follow on other genera and species.
The manuscript is well written, though too compact for some aspects.

Experimental design

It is unclear whether the 79 genomes considered were reannotated in this study, using anvi’o, or that the authors have used the gene prediction and annotation as available in the public databases. The latter is not favorable, as one would rely on predictions and annotations most likely performed within a long period of time, thus relying on different information from public databases. Also, when using genome annotation information for a comparative genomics study, one need to be sure about the quality of the genome annotations, so one should take additional, expert steps beyond the automatic pipelines, to assure that the comparative genomics study starts from a high-quality data set. So the authors are asked to detail what they have done regarding genome annotation.

The authors have searched for secondary metabolites, and found multiple strains with gene clusters involved in the production of gassericin T and gassericin S. More detailed information might be useful to mention in the manuscript, for example the percentages identity and similarity at amino acid level, so not having it as supplemental information.

Validity of the findings

In line 240, the authors state that they did not find a candidate gene marker to distinguish between the two species. However, this seems in conflict with the results reported in line 142, two gene are mentioned to be in present in L. paragasseri and not present in L. gasseri, namely one encoding a phosphatidylserine decarboxylase and one encoding a beta-galactosidase. Please comment on this apparent contradiction.

Additional comments

Minor/textual comments
line 147: paragasseri is not written in italic.

---

## Round 0.2 · accepted · Accept

Thank you for addressing the reviewers' comments and indicating what your analysis adds to the previous analysis done on fewer genomes.